# Use of *Alteromonas* sp. Ni1-LEM Supernatant as a Cleaning Agent for Reverse-Osmosis Membranes (ROMs) from a Desalination Plant in Northern Chile Affected by Biofouling

**DOI:** 10.3390/membranes13050454

**Published:** 2023-04-22

**Authors:** Hernán Vera-Villalobos, Carlos Riquelme, Fernando Silva-Aciares

**Affiliations:** 1Centro de Bioinnovación, Facultad de Ciencias del mar y Recursos Biológicos, Universidad de Antofagasta, Avenida Universidad de Antofagasta, Antofagasta 1240000, Chile; hernan.vera@uantof.cl (H.V.-V.); carlos.riquelme@uantof.cl (C.R.); 2Departamento de Biotecnología, Universidad de Antofagasta, Avenida Universidad de Antofagasta, Antofagasta 1240000, Chile

**Keywords:** *Alteromonas* sp., antifouling, biofouling, reverse osmosis membranes, desalination plant

## Abstract

Biofouling refers to the undesirable growth of microorganisms on water-submerged surfaces. Microfouling, the initial state of biofouling, is characterized by aggregates of microbial cells enclosed in a matrix of extracellular polymeric substances (EPSs). In seawater desalination plants, filtration systems, such as reverse-osmosis membranes (ROMs), are affected by microfouling, which decreases their efficiency in obtaining permeate water. The existing chemical and physical treatments are expensive and ineffective; therefore, controlling microfouling on ROMs is a considerable challenge. Thus, new approaches are necessary to improve the current ROM cleaning treatments. This study demonstrates the application of *Alteromonas* sp. Ni1-LEM supernatant as a cleaning agent for ROMs in a desalination seawater plant in northern Chile (Aguas Antofagasta S.A.), which is responsible for supplying drinking water to the city of Antofagasta. ROMs treated with *Altermonas* sp. Ni1-LEM supernatant exhibited statistically significant results (*p* < 0.05) in terms of seawater permeability (Pi), permeability recovery (PR), and the conductivity of permeated water compared with control biofouling ROMs and those treated with the chemical cleaning protocol applied by the Aguas Antofagasta S.A. desalination plant.

## 1. Introduction

Biofouling is the undesirable growth of microorganisms on water-submerged equipment, infrastructure, or surfaces [1,2]. The initial state of biofouling, known as microfouling, is characterized by aggregates of microbial cells enclosed in a matrix of extracellular polymeric substances (EPSs) [3,4]. EPSs function as a primary settlement for bacteria and diatoms, consequently leading to the colonization of other biofouling components, such as macroalgae spores and invertebrate larvae [4]. Depending on the affected industry (such as aquaculture or desalination plants for water consumption), biofouling can severely affect important operations, resulting in physical damage, biological competitiveness, cultivation system deformation, and environmental modifications, leading to higher economic costs [5].

Biofouling is one of the major problems affecting seawater desalination plants. Biofouling negatively affects the water quality of filtration system products, such as reverse-osmosis membranes (ROMs), through biofilm-enhanced concentration polarization (BECP) and enhanced solute passage, resulting in an increased number of pretreatment steps [6]. Maddah and Chogle (2016) [7] indicated that the primary consequences of biofouling in RO systems are decreased membrane flux and biodegradation, increased salt passage and differential pressure and feed pressure, higher energy requirements, more-frequent chemical cleaning, and a severe decline in permeate quality, thus increasing production costs.

Since the existing chemical and physical treatments are expensive and ineffective, controlling microfouling on ROMs is a considerable challenge. It damages ROM structures, thus producing mechanical interference and a decrease in the permeability of seawater [8,9]. According to Goh et al. (2018) [10], mitigating biofouling in the desalination industry worldwide would cost more than USD 15 billion annually. These adverse effects and the high cost of chemical treatments have resulted in the use of biological molecules as a novel strategy to prevent biofilm formation [11,12] compared with the current antifouling approaches, which are expensive and toxic to the immediate environment and treated surfaces.

Biological approaches have been proposed to solve this problem, such as the use of molecules with quorum-quenching activity, isolated enzymes, and the use of bacterial supernatants with mixes of enzymatic activities [13,14,15]. Quorum quenching aims to inhibit communication between bacteria, preventing their settlement, which generates fouling [15,16]; on the other hand, the use of isolated enzymes or bacterial supernatants is intended to be applied as a ROM-cleaning treatment, either by eliminating the bacterial component [17,18] or degrading the EPS that allows biological settlement [19]. Among these enzymes, those with high potential for use are: alginate lyase, which is capable of degrading EPS polysaccharides, mitigating biofouling formation [20], and xanthine oxidase, which has the ability to produce superoxide anions [21]. On the other hand, enzyme mixes, such as trypsin-EDTA, proteinase K, α-amylase, β-mannosidase, and alginate lyase, have been proposed as an alternative to degrade different biofouling components [22]. However, many of these approximations have only been evaluated on a laboratory scale, which does not allow a real estimation of their performance under conditions of high pressure, water flow, and temperature changes, which are characteristic of fully operational desalination plants [19,23].

Thus, if the use of these new technologies is effective, they could represent an eco-friendly product that decreases the operating costs of the cleaning procedures associated with ROMs maintenance [7]. In this context, the bacterial strain *Alteromonas* sp. Ni1-LEM [24], which produces a thermostable, extracellular, peptide-like compound, is highly significant. It exhibits inhibitory effects on the adhesion capabilities of certain planktonic species and marine benthic microorganisms involved in biofouling [24,25,26,27]. The previous results obtained by our group corroborate the finding that the antifouling compounds secreted by the marine bacteria *Alteromonas* sp. Ni1-LEM disrupt the matrix of the extracellular polymeric substances (EPSs), thus modifying the microbial communities observed in ROMs on a laboratory scale [26,27]. However, experiments performed under real conditions are necessary to corroborate the *Alteromonas* sp. Ni1-LEM supernatant’s cleaning effect on ROMs exposed to the pressure, temperature, and flux conditions that are normally used in reverse-osmosis plants. In the present study, we evaluated the efficacy of the supernatant obtained from *Alteromonas* sp. Ni1-LEM as a cleaning treatment for ROMs using the same conditions applied by Aguas Antofagasta S.A., a company dedicated to supplying drinking water through reverse-osmosis technology in the city of Antofagasta, Chile. The results show that the application of the supernatant obtained from *Alteromonas* sp. Ni1-LEM in the ROMs allows the considerable recovery of permeate production capacity, restoring the topography of the membranes under the pressure and flow conditions used in the seawater desalination industry.

## 2. Materials and Methods

### 2.1. Culture Conditions of Alteromonas sp. Ni1-LEM and Supernatant Production

*Alteromonas* sp. Ni1-LEM [25] was grown for 4 days at 20 °C in a semi-continuous 5 L fermenter with M9 minimal saline medium (casamino acids 1 g/L, Na_2_HPO_4_ 6 g/L, KH_2_PO_4_ 3 g/L, NH_4_Cl 1 g/L, NaCl 21 g/L, 0.1 M MgSO_4_·7H_2_O 10 mL, 0.01 M CaCl_2_·2H_2_O 10 mL, vitamin B1 (1%) 0.2 mL, and glucose (20%) 20 mL) until the stationary phase was reached. Afterward, the culture was centrifuged at 8500 rpm at 4 °C for 15 min. The supernatant was filtered in a tangential flow filtration system (Sartorius AG, Goettingen, Germany) with a 0.2 µm pore size (Hydrosart membrane, Sartorius AG) and stored at 4 °C until further use. For ROM cleaning treatments, the total protein concentration was determined using the Pierce BCA protein assay kit (Thermo scientific, Waltham, MA, USA).

### 2.2. Reverse Osmosis Membrane (ROM) Selection

A ROM model SWC6-LD Hydranautics system with 3.5 years of continuous operation was used in all experiments. The SWC6-LD is a seawater reverse-osmosis membrane manufactured by Hydranautics made of composite polyamide polymer and a low-fouling spiral-wound configuration with dimensions of 40 × 8 inches, and a performance of 12,000 gallons/day, 1200 PSI max pressure, <0.1 PPM of max chlorine concentration, and 45 °C of max operating temperature. ROMs were selected on the basis of the first membranes located in the RO system (Figure 1). The selection of these ROMs was made on the basis of the high biological presence of microfouling according to the operation protocols of Aguas Antofagasta, S.A. Hence, the mentioned membranes in the RO system were replaced by brand new ROMs once their lifetime expired (3.5 years). Aguas Antofagasta SA is a company in charge of purifying water in the town of Antofagasta, Chile. The desalination plant has an installed capacity of 1056 L per second, with an estimation of permeate recovery of 43–45%. To evaluate the effects of the *Alteromonas* sp. Ni1-LEM supernatant as a cleaning agent for ROMs, 42 cm^2^ sections were obtained from 3.5-year-old ROMs (SWC6-LD, Hydranautics, Oceanside, CA, USA) with homogeneous microfouling and kept in sterilized seawater at 4 °C until use. The brand-new ROM (SWC6-LD, Hydranautics) was used as a clean control.

### 2.3. ROM Cleaning Treatments

The cleaning protocol applied by Aguas Antofagasta S.A. was used with certain modifications (varying cleaning solutions), and it was escalated to the Sterlitech CF042 system. This consisted of the passage of different cleaning solutions for a total estimated time of 10 h at 58 psi (4 bar), with recirculation and rest cycles using the following steps:

First Rinse Step: Permeate water was recirculated for 15 min at 58 psi to eliminate the remaining salinity from ROM operations. Next, a solution of 0.1% ethylenediaminetetraacetic acid (EDTA), pH 12, at 35 °C was recirculated for 5 min to eliminate salt precipitates.

Wash Step 1: The membranes were washed with an alkaline solution (sodium tripolyphosphate 0.67 *w*/*v*, EDTA 0.6% *w*/*v*, pH 12) at 35 °C, and 4 cycles consisting of 30 min of recirculation at 58 psi and 45 min of incubation were applied. After every cycle, the alkaline wash solution was replaced with a fresh solution with the same properties.

Second Rinse Step: After the alkaline wash step, ROMs were rinsed with permeate water for 15 min at 58 psi to eliminate the remnants from the previously applied alkaline solution.

Wash Step 1: The membranes were washed with an acid solution (1% citric acid *w*/*v*) at room temperature. In total, 3 cycles consisting of 30 min of recirculation at 58 psi and 45 min of incubation were applied. After every cycle, the acid wash solution was replaced with a fresh solution with the same properties.

Third Rinse Step: The last wash step was applied before assessing the performance. ROMs were rinsed with permeate water for 15 min at 58 psi to eliminate the remnants of the acid solution.

To evaluate the effects of the *Alteromonas* sp. Ni1-LEM supernatant as a cleaning agent for ROMs, treatments replacing alkali, acid, or both solutions with the supernatant (at a final concentration of 200 µg/mL of proteins in the supernatant) were applied and are listed in Table 1.

### 2.4. ROMs Performance Evaluation

ROMs previously cleaned with the respective treatments were placed in two cells of a CF042D crossflow RO system (Sterlitech, Auburn, WA, USA), and their performance was assessed for 15 h of continuous operation. To perform these experiments, pre-osmosis water from the Aguas Antofagasta operation plant was used (pH 7.69, SDI < 5, and 52,850 µS/cm of conductivity), and the operation conditions were similar to those applied by Aguas Antofagasta S.A. in their routine applications (water flux speed: 37.49 cm × seg^−1^, pressure: 825 psi, intake water temperature: 19–21 °C, water flux rate: 2 L/min).

To calculate seawater permeability (*Pi*), the permeate was obtained in a glass container positioned on a digital scale (IND131, Metler Toledo, Toledo, OH, USA), and the difference in the weight was monitored every hour for a total time of 15 h. Equation (1) was applied according to Nagaraj et al.’s work (2017) [11]:(1)$$Pi=\frac{V}{A*t*∆P,}$$
where *Pi* is water permeability (L·m^−2^·h^−1^·bar^−1^), *V* is the volume in liters of the permeate obtained under constant pressure, *P* is the operating pressure (bar), *t* is time (h), and *A* is active ROM area (m^2^).

The average *Pi* values obtained during the experiments were used to estimate the permeability ratio according to Equation (2).
(2)$$Permeability\;ratio\;(PR)=\frac{Pi}{Pcontrol}$$
where *Pi* is the permeability after the respective cleaning treatments and *Pcontrol* is the control related to a clean membrane (ROM-CT).

In addition, to observe membrane integrity and permeate water quality, conductivity (µS × cm^−1^) and pH were measured every hour in the pre-osmosis and permeate water during all experiments.

### 2.5. Evaluation of ROM Cleaning Treatments through AFM Microscopy

ROM sections of 1 × 1 cm that had been previously treated via the cleaning process mentioned above were evaluated using a model Raman-AFM Alpha300 atomic force microscope (AFM) coupled to a model DU970N-BV (EMCCD) CCD camera, and pictures from a 100 µm^2^ surface scan were taken to observe physical modifications.

### 2.6. Statistical Analysis

The values were expressed as mean ± standard error (ES) using Microsoft Excel (Microsoft Corporation, Seattle, WA, USA). The effect of *Alteromonas* sp. Ni1 LEM supernatant as a cleaning treatment was studied via one-way analysis of variance (ANOVA) using STATGRAPHICS Centurion XV Professional (Statgraphics Technologies Inc., The Plains, VA, USA). The means were compared using multiple comparison tests of least significant differences (LSD; *p* ≤ 0.05).

## 3. Results

### 3.1. Water Permeability after Cleaning Treatments

The results regarding the water permeability (*Pi*) over 15 h from ROMs treated with the different cleaning treatments (Table 1) used by the Aguas Antofagasta S.A. desalination plant are shown in Table 2. As expected, the highest ROMs seawater permeability for each hour sampled during the entire bioassay period (15 h) was obtained under the ROM-CT condition, as this was a brand-new ROM without exposure to biofouling (*p* < 0.05). The other treatments, in order of decreasing seawater permeability, were as follows: ROM-SN, ROM-QT, ROM-SN-QT_ac_, ROM-QT_Al_-SN, ROM-M9, and fouled membranes without cleaning treatments (ROM-BF). In addition, the ROM-M9 treatment, corresponding to the bacterial culture medium containing *Alteromonas* sp. Ni1-LEM, was used. This treatment was carried out without the antifouling bioactive substance and exerted no antifouling effect; no significant differences were found (*p* < 0.05) compared with the ROM-BF condition. In contrast, when water permeability was compared between different ROM cleaning treatments, the results showed that during the 15 h of the experiment, the highest permeability was obtained in the ROM-SN treatment, a condition that corresponded to a cleaning procedure using only the bacterial supernatant (Table 2).

### 3.2. Permeability Ratios of Different Cleaning Treatments

The results of the permeability ratio (PR) measurements conducted over 15 h for ROMs treated with different cleaning treatments are shown in Table 3. The permeability ratio (PR) of each ROM was analyzed after cleaning treatments using ROM-CT as a reference (brand new ROM). The results showed that the treatment with the highest PR was the bacterial supernatant (ROM-SN) with a PR of 0.7205, followed, in decreasing order, by ROM-SN-QT_ac_, ROM-QT, ROM-QT_Al_-SN, ROM-M9, and ROM-BF, with values of 0.6926, 0.6873, 0.6866, 0.5390, and 0.5310, respectively.

As observed in relation to the (Pi) values, when PR was compared between different ROM cleaning treatments, the PR results obtained after 15 h of experiments showed that the highest recovery was achieved by ROMs treated with the bacterial supernatant ROM-SN as a cleaning treatment (*p* < 0.05; Table 3). In contrast, no significant differences were observed between the ROM-BF and ROM-M9 treatments (*p* > 0.05).

### 3.3. Permeate Water Conductivity and pH Values after Reverse-Osmosis Processing Using ROMs from Different Cleaning Treatments

The results of permeate water conductivity (µS × cm^−1^) obtained after 15 h of the ROMs experiment and those previously cleaned with different treatments are shown in Table 4. The results showed that the highest conductivity of permeate water was obtained in the ROM-QT_Al_-SN treatment, followed, in decreasing order, with significant differences (*p* < 0.05), by ROM-BF, ROM-QT, ROM-SN-QT_ac_, ROM-M9, ROM-SN, and finally ROM-CT. This tendency was observed in each hour sampled throughout the experiment. No significant differences were observed in pH values (*p* > 0.05) between ROM cleaning treatments. However, a decrease in pH was observed in the permeated water in all treatments compared with the pH observed in feed water (pre-osmosis), with an average value of 6.93 ± 0.04.

### 3.4. Supernatant from Alteromonas sp. Ni1-LEM Restored Qualitative Aspects in ROMs from the Desalination Plant

To observe how cleaning treatments modified the physical aspects of ROMs, the ROM surface was analyzed through atomic force microscopy (AFM). The results showed that ROM-CT (brand new membranes) samples had a rough texture, highlighting the presence of small regular peaks on their surface (Figure 2A). However, when ROMs were exposed to continuous work for 3.5 years, the presence of biofouling drastically modified the ROMs’ surface to a mostly “spongy” appearance (Figure 2B). Nevertheless, when membranes were submitted to a cleaning treatment, the control topography was partially recovered. A considerable level of recovery was observed in membranes treated with the *Alteromonas* sp. Ni1-LEM supernatant (ROM-SN), either as a single treatment (Figure 2E) or mixed with chemical components (Figure 2F). This result indicates that the *Alteromonas* sp. Ni1-LEM supernatant effects were maintained under operating conditions (high pressure and continuous flow).

## 4. Discussion

Researchers have long searched for novel economical and eco-friendly methodologies for cleaning ROMs, and diverse approaches have been evaluated in an attempt to achieve this goal. Several authors have proposed modifications in ROMs structures to improve their chemical and physical properties [28], as well as the application of different chemical or biological components as cleaning agents with less toxic effects [11,12]. In this context, the use of microorganisms that can degrade the biofouling EPS matrix has exhibited considerable promise [27,29,30]. However, several of these approaches have been evaluated only under laboratory conditions and on a small scale—conditions that, in many aspects, are not similar to those observed in the routine operation of desalination plants. These differences have been primarily noted in water flow rates, operation pressure, and the temperature to which ROMs are exposed during daily cleaning treatments at desalination plants. These factors cannot be underestimated because they affect the performance of these promising approaches that have been observed in the laboratory. In this study, we used the bacterial supernatant of the *Alteromonas* sp. *Ni-LEM* strain as a cleaning agent under real-life operating conditions according to the protocols used by a desalination plant in charge of supplying drinking water to the city of Antofagasta, Chile (Aguas Antofagasta S.A.).

The results showed that the Ni1-LEM supernatant exhibited considerable potential as a cleaning agent under real-life conditions. The data showed that the Ni1-LEM supernatant resulted in the best outcomes in relation to water permeability, the recovery of permeability (PR), and conductivity among all analyzed treatments, and it reached the closest values to the control conditions (ROM-CT).

The use of the *Alteromonas* sp. *Ni1-LEM* supernatant also resulted in improved permeate quality according to the conductivity values. Lower permeated water conductivity implies a higher retention of salts by ROMs [19,31,32], meaning that the ROM-CT membrane retained more salts from the feed water (feedwater conductivity: 52,850 µS × cm^−1^), resulting in permeate water with fewer salts compared with the other treatments. However, the permeated water conductivity from the ROM-CT treatment was used as a reference; thus, treatments with conductivity values near to those of ROM-CT showed better cleaning performance. In this context, the treatment using the bacterial supernatant (ROM-SN), which exhibited lower permeated water conductivity compared with other cleaning treatments, led to better performance when comparing the conditions.

Traditional chemical treatments, primarily those that use alkali solutions to increase cleaning efficiency, require an operating temperature from 35 °C to 38 °C, which is reflected in the increased costs associated with cleaning treatments due to energy use (Aguas Antofagasta SA). In contrast, although the *Alteromonas* sp. *Ni1-LEM* supernatant is a temperature-sensitive solution [25], this did not pose a problem because the best results observed with *Alteromonas* sp. supernatant on a laboratory scale (without pressure and water flux) were obtained at room temperature [27], a condition also observed in the present study compared with other treatments (Table 2 and Table 3). This could be reflected in a decrease in operating costs due to the application of the supernatant. Several authors have reported that traditional chemical treatments, in addition to eliminating biofouling, generate adverse effects on ROM structures, reducing their operation life [9,33,34]. The *Alteromonas* sp. Ni1-LEM supernatant only interacted with biofouling components, such as the formed EPS, and could thus be successfully used against a broad spectrum of microalgal biofilms without negatively affecting the ROM structures [26,27].

Atomic force microscopy (AFM) provides several tools with which to evaluate the physical and micromechanical properties of samples. This is achieved through the interaction of a cantilever probe with the surface to be studied [35]. Thus, AFM has previously been used in RO systems to evaluate physical changes on the surfaces of ROMs [36,37,38,39,40], with the performance of ROMs considerably depending on their physical–chemical properties, which decrease in the presence of biofouling. A previous study by [41] indicated that biofouling produces topographical differences in ROMs, demonstrating that this biological event has adverse effects on desalination systems and their permeate production processes. Our results showed that biofouling generated modifications on ROM surfaces exposed to continuous operation for 3.5 years (Figure 2B), as well as those undergoing periodic cleaning treatments that are carried out with greater intensity when HABs (harmful algae blooms) appear; for example, the bay of Antofagasta suffered from HABs events in 2019 and 2020, and these events are the cause of an increase in the cost production of potable water and a decrease in the average life of ROMs [42]—these modifications are reflected in a decrease in permeate volume (Table 2). However, when the *Altermonas* sp. Ni1-LEM supernatant was used as a cleaning treatment, a recovery in terms of the topography was observed (similar to the control samples: ROM-CT). This recovery was not observed when chemical treatments were used as cleaning methods. The topography observed in the ROM-SN treatment was also correlated with the results obtained regarding the comparison of permeate volumes, because the ROM-SN treatment (treatment with the *Altermonas* sp. Ni1-LEM supernatant) had the highest performance compared with the other chemical treatments. These data support the idea that the *Alteromonas* sp. Ni1-LEM supernatant can be applied in operating conditions as an eco-friendly alternative for ROM cleaning in the desalination industry.

One of the biggest bottlenecks preventing the industrial application of these new eco-friendly approaches, allowing them to compete with more traditional treatments, is the scaling of these processes for higher-volume production [29]. In this context, our group previously scaled and produced a massive 400 L culture of *Alteromonas* sp. Ni1-LEM to obtain and evaluate the production of a supernatant with antifouling effects. The data showed that this massive culture had a supernatant production yield with a concentration of 614 µg mL^−1^ of protein. As mentioned above, the application of the supernatant as an ROM-cleaning treatment at a concentration of 200 µg mL^−1^ of protein as a working solution indicated that the massive culture generated a 3× concentrate of the working solution.

Additionally, preliminary economic studies showed that the production cost of 1 L of working solution of supernatant (200 µg mL^−1^) has an approximate cost between USD 0.14 to 0.19, which would allow generating a business model for its large-scale production.

## 5. Conclusions

The results show that the *Alteromonas* sp. Ni-LEM supernatant is a promising substance with a high potential for industrial application because it can be applied under the conditions of pressure, water flow, and temperature normally used in seawater desalination plants. Future work will focus on the application of a semi-industrial-scale Ni1-LEM supernatant based on technological transference projects.

## Figures and Tables

**Figure 1 membranes-13-00454-f001:**
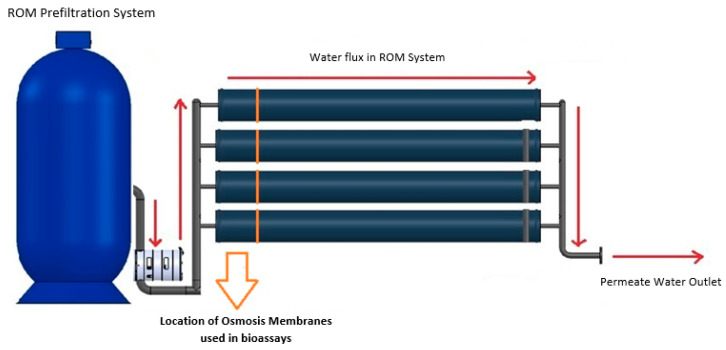
Desalination system scheme. Overview scheme of the desalination system used by Aguas Antofagasta SA. Red arrows denote the water flux inside the RO system to obtain permeated water. Orange arrow indicates the first ROMs exposed to water in the reverse-osmosis system and ROMs used in all the bioassays.

**Figure 2 membranes-13-00454-f002:**
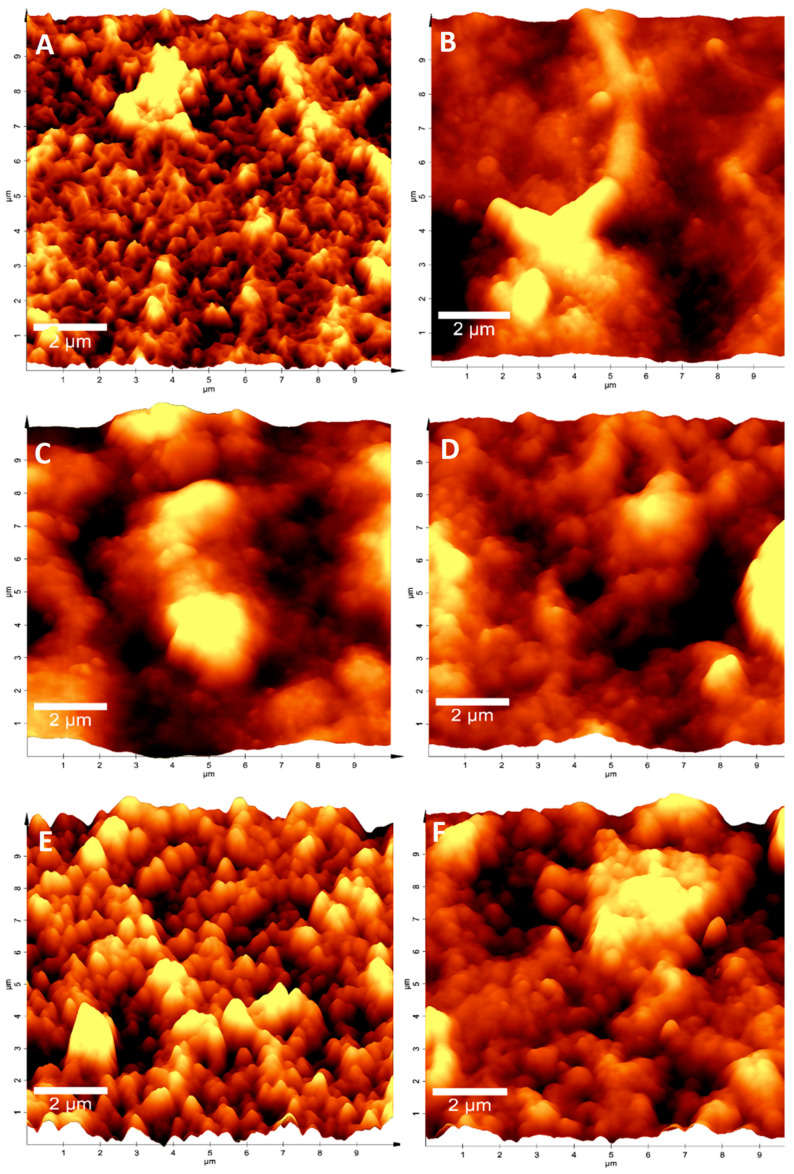
ROMs AFM microscopy after cleaning processes. Imaging of ROMs surface obtained through AFM microscopy: (**A**) brand new membrane ROM-CT; (**B**) treated with M9 ROM-M9; (**C**) fouled-membrane ROM-BF; (**D**) chemical treatment ROM-QT; (**E**) Ni1-LEM treatment ROM-SN; and (**F**) Mix ROM QT_al_-SN. The results showed that Ni1-LEM treatment considerably restored qualitative aspects of ROMs.

**Table 1 membranes-13-00454-t001:** Description of ROMs treatments. All ROMs were model SWC6-LD Hydranautics, and fouled ROMs were obtained after 3.5 years of continuous operation. ROMs were subjected to a cleaning process described by Aguas Antofagasta S.A., modifying chemical mixes according to treatments. Additionally, Fouled ROMs (ROM-BF) and Unfouled ROMs (ROM-CT) without cleaning treatments were used as controls.

Treatment Abbreviation	1st Rinse Step (58 psi -15 min)	Wash Step 1 (58 psi) (R: 30 min I: 15 min) X4	2nd Rinse Step (58 psi -15 min)	Wash Step 2 (58 psi) (R: 30 min I: 15 min) X4	3rd Rinse Step (58 psi -15 min)
ROM-M9	R: 0.1% EDTA pH 12 at 35 °C	M9 media at RT	R: permeate water at RT	M9 media at RT	R: permeate water at RT
ROM-QT	R: 0.1% EDTA pH 12 at 35 °C	STPP 0.67 *w*/*v*, EDTA 0.6% *w*/*v* pH 12 at 35 °C	R: permeate water at RT	1% citric acid *w*/*v* at RT	R: permeate water at RT
ROM-SN	R: 0.1% EDTA pH 12 at 35 °C	*Alteromonas sp.* Ni1-LEM supernatant at RT	R: permeate water at RT	*Alteromonas sp.* Ni1-LEM supernatant at RT	R: permeate water at RT
ROM-QT_Al_-SN	R: 0.1% EDTA pH 12 at 35 °C	STPP 0.67 *w*/*v*, EDTA 0.6% *w*/*v* pH 12 at 35 °C	R: permeate water at RT	*Alteromonas sp.* Ni1-LEM supernatant at RT	R: permeate water at RT
ROM-SN-QT_ac_	R: 0.1% EDTA pH 12 at 35 °C	*Alteromonas sp*. Ni1-LEM supernatant at RT	R: permeate water at RT	1% citric acid *w*/*v* at RT	R: permeate water at RT
ROM-BF	Fouled reverse osmosis membrane without cleaning treatment				
ROM-CT	Brand new reverse osmosis membrane without cleaning treatment				

R: Recirculation. I: Incubation. RT: Room Temperature.

**Table 2 membranes-13-00454-t002:** Comparison of water permeability (Pi) from ROMs treated with different cleaning protocols. ROM-CT (Clean ROM); ROM-BF (Fouled ROM without treatment); ROM-M9 (M9 culture media); ROM-QT (Complete Chemical Treatment); ROM-SN-QT_Ac_ (*Alteromonas* sp. Ni1-LEM supernatant + acid chemical treatment); ROM-QT_Al_-SN (*Alteromonas* sp. Ni1-LEM supernatant + alkali chemical treatment); and ROM-SN (Only with *Alteromonas* sp. Ni1-LEM supernatant). Statistical differences (*) against ROM-BF are from four replicates and validated with one-way ANOVA LSD (*p* < 0.05).

Treatments	Average Pi (L·m^−2^·h^−1^·bar^−1^)	ED
ROM-CT	0.5272 *	±0.0016
ROM-SN	0.3747 *	±0.0026
ROM-SN-QT_ac_	0.3642 *	±0.0014
ROM-QT	0.3625 *	±0.0044
ROM-QT_Al_-SN	0.3620 *	±0.0017
ROM-M9	0.2765	±0.0046
ROM-BF	0.2733	±0.0051

**Table 3 membranes-13-00454-t003:** ROM permeability recovery (PR) of ROMs after treatments with different cleaning protocols. ROM-CT (Clean ROM); ROM-BF (Fouled ROM without treatment); ROM-M9 (M9 culture media); ROM-QT (Complete Chemical Treatment); ROM-SN-QT_Ac_ (*Alteromonas* sp. Ni1-LEM supernatant + acid chemical treatment); ROM-QT_Al_-SN (*Alteromonas* sp. Ni1-LEM supernatant + alkali chemical treatment); and ROM-SN (Only with *Alteromonas* sp. Ni1-LEM supernatant). Results showed that highest PR was reached by ROMs treated with *Alteromonas* sp. Ni1-LEM supernatant. Statistical differences (*) against ROM-BF are from four replicates and validated with one-way ANOVA LSD (*p* < 0.05).

Treatments	Water Permeability Recovery (PR)	ED
ROM-CT	1.0 *	0
ROM-SN	0.7205 *	±0.0038
ROM-SN-QT_Ac_	0.6926 *	±0.0046
ROM-QT	0.6873 *	±0.0064
ROM-QT_Al_-SN	0.6866 *	±0.0039
ROM-M9	0.5390	±0.0048
ROM-BF	0.5310	±0.0081

**Table 4 membranes-13-00454-t004:** Permeate conductivity (µS·cm^−1^) after ROMs treatments with different cleaning protocols. ROM-CT, ROM-BF, ROM-M9, ROM-QT, ROM-SN-QT_Ac_, ROM-QT_Al_-SN, and ROM-SN. The 15-h results showed that lower conductivity was achieved by ROMs treated with *Alteromonas* sp. Ni1-LEM supernatant. Statistical differences (*) against ROM-BF are from four replicates and validated with one-way ANOVA LSD (*p* < 0.05).

Treatments	Average Conductivity (µS·cm^−1^)	ED
ROM-CT	434.77 *	±4.64
ROM-SN	469.517 *	±4.26
ROM-M9	479.717 *	±3.34
ROM-SN-QT_Ac_	487.893 *	±2.06
ROM-QT	490.675 *	±2.21
ROM-BF	511.25	±1.49
ROM-QT_Al_-SN	528.253 *	±4.14

## Data Availability

The data presented in this study are available on request from the corresponding author.

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
