# Peer review of "Use of Alteromonas sp. Ni1-LEM Supernatant as a Cleaning Agent for Reverse-Osmosis Membranes (ROMs) from a Desalination Plant in Northern Chile Affected by Biofouling"

_membranes, 2023, doi:10.3390/membranes13050454_

Round 1

Reviewer 1 Report

The manuscript titled “Use of Alteromonas sp. Ni1-LEM supernatant as a cleaning agent for reverse osmosis membranes (ROMs) from a desalination plant in northern Chile affected by biofouling” and written by Hernán Vera Villalobos et al. reports the results of different products used for chemical cleanings of RO membranes under biofouling conditions. The paper is interesting, however, there are some drawbacks that should be addressed by the authors. I recommend a minor revision based on the following comments:

1.      Page 1, line 24. The units are not necessary in this abstract.

2.      The introduction should be extended. There are relevant studies related with biofouling in RO desalination plants in the literature that have not been commented. There are some pre-treatments that helps to mitigate biofouling (Membranes 12(12),1209;      Separations 9(1),1), some studies are related with the preparation of new fouling-resistant membranes (Desalination 543,116107; Membranes 12(10),928; Desalination 540,115997; Membranes 12(9),851).

3.      Page 2, line 87, reverse osmosis membranes was already abbreviated, please, use the abbreviations properly. Revise the entire manuscript. Also the title of section 2.2, etc.., in page 5 line 205

4.      Improve the quality od Table 1, it is impossible to be read

5.      Could the authors show the experimental data of the 3.5 years of operation (pressures, flows, conductivities, RO system configuration, number of pressure vessels and ROM element per pressure vessel). This is crucial when the biofouling impact and mitigation is evaluated. 3.5 years is not much for a SWRO desalination plant. Why the ROMS had to be replace so early? What was the chemical cleaning in place protocol in the real desalination plant?.

6.      Table 3 shows a 100% patency recovery for the ROM-CT treatment. After 3.5 years of operation, this would be impossible in industrial scale installations. The authors should comment that these efficiencies would not be comparable to those that could be obtained in large scale plants where spiral wound elements are available. Although it is true that the results are very useful at a comparative level between chemical cleaning treatments.

7.      Section 4 should be separated into a discussion section and conclusions section.

8.      Table 5 is not necessary, it has only 1 line, this results can be written in 1 or 2 lines. Remember the section conclusion should not include results in terms of tables or graphs.

Reviewer 2 Report

Bio fouling is a serious problem as it can significantly influence RO membrane performance. Formation of bio fouling layers in membrane modules greatly depends on conditions of biological growth (such as algae bloom, wastewater pollution of intake water body, etc.). But in all cases the bio growth should be evaluated. The proposed reagent to clean RO membranes and to remove biofilms can be valuable only on a condition that we are aware of the amount of accumulated biofilm and it’s influence on membrane performance. 

In the article authors have performed cleaning of membrane samples taken out of spiral wound modules after 3.5 years of their operation. I should underline that membranes have experienced previously the routine cleanings that enabled them to “survive” throughout 3 years of operation . The experimental cleanings were performed by the authors without any idea of the amount of bio fouling material collected on membrane surface and it’s influence on membrane performance.

Minor editing of English language required

Reviewer 3 Report

The novelty of the work done should be written at the end of the introduction section.

All the specs for the chemicals and instruments used in this work should be provided.

The complete specs of the membranes used in this work from their manufacturer should be provided, e.g., "The SWC6-LD is a seawater reverse osmosis membrane manufactured by Hydranautics. It has a length of 40 or 80 inches, a diameter of 4/8 inch, and weighs xx lbs. The membrane type is xxxxxxxxx with an active membrane area of xxx ft2. The feed spacer is x mil, and the permeate flow rate ranges from xxx GPD/m3 to xxx GPD. The applied pressure ranges from xxx PSI/bar to xxx PSI, with a maximum pressure of xx PSI. The stabilised salt rejection ranges from 99.6% to 99.8%, while the typical boron rejection ranges from 83.0% to 91.0%".

The capacity of the RO plant should be mentioned, as well as how it works and what percentage of the water it recovers.

Also, out of the properties of the seawater, at least the TDS should be mentioned.

"Reverse osmosis membranes were selected on the basis of the first membranes located in the RO system (Figure 1)". Where is Fig. 1?

The quality of Table 1 should be improved.

Since this is experimental work, the CF042D crossflow RO system should be shown in a picture or on a diagram.

Equations are not numbered.

"Where Pi is water permeability (l·m–2 ·h–1 ·bar–1 )". Please unify the units throughout the manuscript, e.g., (l or L).

"V is the volume of the permeate obtained under constant pressure (L)." What is the meaning of "L" here?

Is there any uncertainty analysis carried out during this work?

Please explain how this work is different from the one that was published by the same authors and others and called "Characterization and removal of biofouling from reverse osmosis membranes (ROMs) from a desalination plant in Northern Chile, using Alteromonas sp. Ni1-LEM supernatant" (reference 16 in this work).

Please increase the resolution of Fig. 2. Where the (A, B, C, etc.) should be written on the Fig.

Even though this cleaning method was tested in a lab almost 30 years ago, how much would it cost to use it in the desalination industry?Please discuss/assess.

The conclusion should be a concise section separate from the discussions.

Please recommend future work.

References cited are very old; only two of them are from 2022. Please then update so that 50% are from the past 2 years.

A lot of the citations inside the manuscript are not appropriate and must be corrected according to the instructions for authors.

 Moderate editing of English language and styles should be performed.

Round 2

Reviewer 3 Report

Since the authors responded to most of my comments. The manuscript in its present form is acceptable for publication.

*